# Parental Perceptions of Family-Centred Supports for Children with Developmental Disabilities

**DOI:** 10.3390/ijerph20054205

**Published:** 2023-02-27

**Authors:** Roy McConkey, Pauline O’Hagan, Joanne Corcoran

**Affiliations:** 1Institute of Nursing and Health Research, Ulster University, Belfast BT1 6DN, Northern Ireland, UK; 2Positives Futures, Bangor BT20 5BE, Northern Ireland, UK

**Keywords:** family, care-givers, parents, developmental disabilities, family-centred supports, social isolation, rural, qualitative

## Abstract

Family-centred interventions have long been advocated in paediatric practice and in public health, but their implementation is rarer with children experiencing developmental disabilities. Moreover, the uptake is lower in families from more socially deprived backgrounds. Yet there is robust evidence that such interventions bring benefits to the family caregivers as well as to the affected children. The present study emerged from a support service that had been located in a rural county in Ireland in which nearly 100 families of children with intellectual and developmental disabilities had participated. Using a qualitative research methodology, interviews were conducted with 16 parents who had taken part in the service with the aim of gaining insights into the ways a family-centred service had been of value to them. The themes identified in their responses were validated in two ways. All parents were given the opportunity to provide their perceptions using a self-completion questionnaire, and nearly 50% responded. In addition, seven health and social care staff who had referred families to the project were asked their views through personal interviews. The core theme to emerge was the focus placed on family engagement by the service, with four subthemes emerging: parental confidence boosted; children developed; community connections were made; and supportive staff. These insights should help existing health and social care services to become more family-centred and inform the development of new support services in response to the high levels of unmet needs among marginalized families in even the most affluent countries.

## 1. Introduction

Parents of children with developmental disabilities and chronic illnesses face many challenges that often extend beyond the extra care-giving demands they experience [1,2]. Mothers especially are at greater risk of poor physical health and emotional wellbeing [3], which is further exacerbated by social isolation and poverty [4]. Yet to date, the efforts of many health and social services has been on managing the child’s condition, such as clinic-based therapies, hospital treatments or educational provision. Support for families is often not readily available, even in the most affluent countries, and is non-existent in poorer nations [5,6].

Arguably, a change of ethos is needed in health and social care services, notably a focus on meeting the needs of families as well as those of the child. Indeed, a case has been made for focusing on families within primary, secondary and tertiary public health policy and practice [7]. Likewise, family-centred interventions are promoted within paediatric services [8]. The rationale is similar in both domains. Families have a strong influence on all children’s development and wellbeing, for better or worse. Hence children with additional needs are helped by helping the families. Parents have proven themselves to be effective teachers and therapists for their child when given appropriate guidance and support [9]. Mobilising the efforts of families reduces the need for and dependency on professional services with their attendant costs [10]. Preventative interventions with families can start in early childhood and avoid crises developing that require more specialized and costly ‘tertiary level’ interventions. However, even when such crises are unavoidable, the approach still needs to be family-centred so that the child’s place within the family is maintained [7].

These arguments are equally valid for children who have developmental disabilities. Latterly, they have been endorsed in a major international review of clinical practice in relation to autism, triggered by the growth in its prevalence across the world [11]. The review proposed a stepped model of service responses and noted: “*The role of the family is almost always critical; thus, stepped care/personalised health models must take into account the needs, abilities and ‘personal costs’ (not just financial) to the family and directly to the autistic person*” (p. 23).

To date, the means for realizing these aspirations are largely untested as family-centred interventions have not been widely implemented, although others have long argued for them across other impairment conditions [12]. In particular, the focus has been on the expectations and opinions that researchers and practitioners hold of parents within the context of family-centred interventions as epitomized in the conceptual models that have been proposed [8,13]. By contrast, we also need a better understanding of how parents view such interventions and what they value from them. This is especially so in the case of more marginalized families—those who are less educated, single carers, or unemployed—who are reluctant or unable to seek assistance from statutory services [14].

The present study provided an opportunity to address this gap, albeit within one rural community. The main aim was to document the lived experiences of families who had been involved in an innovative, family-centred support project for children with diagnosed or suspected developmental disabilities so as to discover what they had valued from the experience. Moreover, ideas for better meeting their needs were also sought. This provided insights into parental priorities and desired outcomes that other services across various childhood conditions could build into their planning in order to make them more family-centred.

## 2. Materials and Methods

### 2.1. Brighter Futures Service

Northern Ireland is part of the United Kingdom with a national health and social care service that provides support services to children with disabilities at no cost to families. These providers are referred to as ‘Trusts’. In addition, non-governmental, voluntary services also provide services and may be funded from charitable foundations to do so. In these instances, they accept referrals from Trust staff with an assessment and support role, such as community nurses and social workers. Positive Futures is a leading disability NGO in Northern Ireland, and with funding from the UK Community Fund, launched the Brighter Futures Project in a predominantly rural county in Ireland with a population of around 62,000. The project built upon their experiences of family-centred services in other locations in Northern Ireland [15].

The project’s focus was mainly on children who had received a diagnosis of a developmental disability, but in recent years, this was extended to children who were waiting for assessments, notably for the Autism spectrum. The project staff were recruited and lived locally. Hence, they were familiar with the culture and customs of the area and knew the community facilities that were available. The staff had experience of working with children who had additional needs but they received further training in family support and child development. They visited the family home monthly over a 12-month period with phone calls in between visits. During the COVID-19 lockdowns in 2020/21, contact with families was by phone or through Zoom.

The role of staff was two-fold. They identified in consultation with families, the learning targets that parents could focus on that were suited to the child’s developmental level, and together with families, they devised activities that could take place in the family home or during outings in the local community when the children could participate in leisure and sport activities. These focused mostly on social and communication skills and included activities such as using pictures to communicate choice of drinks, joining in playing Lego with a sister, turn-taking activities in imaginative play, participating in team games and reading social stories about keeping safe. Parents were also concerned to promote their child’s confidence and independence. Here, the activities included dressing oneself for school each morning; sleeping in one’s own bed; accompanying Mum to the supermarket and going swimming with Dad once a week. Managing the children’s behaviour and reducing temper tantrums were another focus, notably, stopping children running off, removing the child when they became anxious, and preparing children for changes.

The second focus of project staff was on the wellbeing of parents and mothers, especially. They provided a listening ear during home visits or in small group meetings; they guided parents to available community resources; they referred parents to other services or sought advice from these personnel that they could pass on to families and they accompanied them to clinic or hospital appointments to assist in exchanging information and understanding the actions to be taken by the services. Furthermore, social activities were organized for mothers, such as beauty sessions, yoga and relaxation. Group sessions allied with sport activities were organized for fathers, and similar activities were provided for siblings. In addition, family activity days, such as swimming, soft play and adventure parks, were arranged to which all family members were invited so that families could meet each other socially as well as obtain ideas that they could repeat in their own time.

During the course of the project, the staff received additional training in assessing children’s development and devising learning activities to accord with the children’s needs and family aspirations. Their counselling skills in working with families were further enhanced through workshops or as they participated in the sessions alongside the resource persons that had been invited to lead them. The staff also shared their experiences and expertise with one another at regular staff meetings and supervision sessions. Further details of the project, alongside details of its impact on the children and parents, are presented in an accompanying paper [16].

### 2.2. Description of the Participants

In all, 96 families have participated in the project thus far, representing 91% of families referred to the project over a five-year period from 2017 to 2022. The uptake rate was high with few drop-outs. Based on local area Multiple Indicators of Social Deprivation for N.I. [17], nearly two-thirds of the families (65%) resided in areas that fell within the top 30% of more deprived areas, with only 3% living in the 30% least deprived areas. Moreover, this measure is thought to underestimate the extent of rural deprivation.

For 70 families (73%), both natural parents resided together and a further two were a reconstituted family (2%), while 24 (25%) were single parents. The median number of children in the household was two (range 1 to 7). In all, 31 (32%) families reported having another child with a disability in the family.

The mother was reported to be the primary carer of the child with special needs in 84 (88%) families. In seven families (7%), both parents were named as carers, and in five families (5%), the father was the primary carer.

The mean age of the primary carer was 39 years (range 22 to 61 years). In all, 44 (46%) primary carers had attended higher education, 6 (6%) left school at 18 years, 30 (31%) had GCSEs and 15 (16%) left school at 16 years with no qualifications.

Nearly two-thirds (n = 60:63%) of the primary carers were not in employment, while 14 (15%) worked full-time, 19 (20%) part-time and two occasionally (2%). Of the 96 families, 46 (48%) were reported to own their own home, and 50 (52%) did not.

#### Characteristics of the Children

Of the children who completed the project, 78 were boys (71%) and 32 (29%) were girls. (Some families had two or three children with disabilities referred to the project over the five-year period). Their median age when starting the project was 6.5 years (range 9 months to 13 years). In all, 62 (56%) were reported to have autism, 33 (30%) had a learning disability and 29 (26%) had other developmental disabilities. In addition, 18 (16%) children had other conditions mentioned. Note that children could have more than one condition recorded. A further 18 (16%) children were awaiting a diagnosis.

The children attended the following facilities: 49 (31%) mainstream schools; 41 (37%) special schools or units; 19 (17%) preschools; and 7 (6%) were too young to attend school.

### 2.3. Evaluation Methods

In order to gain insights into the parents’ lived experiences of the project, qualitative methods were used informed by interpretative phenomenological analysis approaches [18]. A university-based researcher (RM) who was uninvolved with the project was commissioned to act as the independent evaluator. Information was sought also from health and social services staff who had referred families to the project and had ongoing contact with them. This was intended to provide some external validation of the parental experiences.

Purposive sampling was used by the Project Manager to identify 19 parents that represented the diversity of families involved with the project over the five years [19]. Three parents were not contactable but 16 parents who had completed the project across different years consented to take part in a telephone interview with the independent evaluator and assurances of confidentiality were given (84% response). The interviews were audio-recorded and lasted on average 20 min, resulting in over five hours of recordings.

In addition, at the end of their 12-month period of home visits, parents were requested to self-complete anonymously, a brief questionnaire that summarised their experiences with the project. In all, 49 questionnaires were returned to the independent evaluator (52% response).

In the questionnaire and interviews, the parents were asked to comment on what they felt were the positive aspects of the service, how it might have made life better for them as parents, any changes they had seen in the children and in the family, and ideas for how the service might be improved. Additional questions were asked of parents who had left the project up to a year or more before, including how they had managed since and their perceptions of any lasting impact the project had on the child and family. During the interviews, additional questions were used to elicit richer insights from the parents about their experiences with and reactions to the project.

In addition, telephone interviews were also conducted with seven health and social service staff who had referred families to the project. This consisted of four social workers, two community nurses (health visitors) and a member of staff from a family centre. All were female and had been in their present post from two to 20 plus years. These interviews also probed the wider learning that the project had for other health and social care support services to children and families.

#### Data Analysis

The parental interviews were transcribed verbatim and parental responses to the self-completion questionnaires were listed verbatim in a Word document. A thematic content analysis was then undertaken across all the questionnaire and interview transcripts using the six steps proposed by Braun and Clarke [20]. The subthemes were initially identified by the independent evaluator (RM), but these were then cross-checked by the two other authors with responsibility for managing the project. It was evident that data saturation had been achieved as no new themes emerged in the analysis of later transcripts.

As is common practice in qualitative research, verbatim quotes are used to provide readers with the actual words used by the respondents who tended to write shorter written responses than did the interviewees. We also use the quotes to illustrate the range of experiences that parents reported under the subthemes, notwithstanding that other parents had also provided similar quotes.

A similar process was followed in analysing the transcripts from the interviews conducted with the referring staff. This provided further validation of the main theme and subthemes, although these informants provided more information around the improvements to the project than did parents, as well as its implications for other health and social support services to families.

A form of member checking was undertaken when the identified core theme and subthemes were presented at a dissemination event organized by the project and attended by over 40 parents and service personnel. Through questions and discussions, the identified themes were confirmed by attendees.

## 3. Findings

One core theme—family engagement—was identified with four interacting subthemes emerging from this. These are summarised in Figure 1 and described in this section using verbatim quotations from parents identified by a number. A further section summarises the improvements to the project that informants had suggested.

### 3.1. Family Engagement

The engagement with the family as a whole was a dominant theme across all the responses. It was not just the children who were the focus of attention but the needs of parents and siblings were also addressed. The home-based visits to the family home were seen as central to placing the focus on families. Partnerships with families were built through the engagement of the parents in selecting learning targets, in making plans for the child and for the other family members, followed by review meetings held at three-monthly intervals. Some representative responses concerning the theme of family engagement are as follows:

*It has made us as parents consider and reflect on what we value as important for our child and how Brighter Futures can support us to achieve this* (1).

*Brighter Futures listens to what we would like to do and they make it happen* (43).

*Regular review meetings to discuss the whole family and any issues that arise* (13).

*The support the (other) kids have received has made it easier to manage their behaviour and helped them socially and emotionally* (40).

*The wider family circle see (Name) now as more sociable* (33).

The engagement with the family was also identified by the staff who had referred children to the project:

*The support they provide is to the family and they look at the family as a whole. Few services do this* (Social Worker).

Engaging the family in selecting learning targets for the child brought wider gains for the family.

*Communication is better (in the family). We have more respect for each other* (12).

*Mammy, daddy and brother have a more settled life because of the changes he has been through* (30).

The family also benefited when staff spent time with the child at home or when the children were taken to community activities. This meant that parents could focus on their other children or take a break for themselves:

*Brighter Futures gives us time as a family to do “normal” family things. Time to focus on each other and time for ourselves as individuals* (24).

*I got to spend time with my other daughter … we got out together or were able to do her homework uninterrupted. She was needing individual attention as she was reacting to the fact that my other child got attention* (2).

*His involvement in the summer scheme, I feel has helped us as a family to achieve a more successful camping trip* (19).

Parents went on to describe the outworking of their experience of family engagement. Four sub-themes were evident in their responses from parents, as shown in Figure 1.

### 3.2. Confident Parents

Parents often mentioned feeling more confident as parents as a result of their involvement with the project:

*Through the project I have become much better informed of her condition and what coping mechanisms I can employ. I could not have arrived at this point without their guidance* (24).

*More confident in asking for help. We are more accepting of his condition. We’ve had respite time for us as a couple and a chance to refresh* (4).

*Gave me more confidence to try new things. I have enjoyed connecting with other parents at parent group* (6).

*I am more confident. I have found my voice to speak up for my child. I tackle a problem head-on* (96).

The emotional support offered by the project was especially valued.

*I was going through a stressful time. It has helped me cope better. I feel in a much better place now* (2).

*I can now face problems with a stronger mind-set; the future is brighter* (11).

*It gave me support and strength and advice on looking after my mental health* (02).

*Gave me support when I was feeling there was no hope* (03).

*Short-term respite has helped with my emotional wellbeing* (20).

Likewise, the information and guidance provided by the project staff was commented on:

*They advised of services I was not aware of* (17).

*Helped me find a programme in local college and I completed English, Maths and First aid etc. I am now going to do a child care course* (19).

Building the confidence of parents was also confirmed by the staff who had referred the families.

*They give families resilience, help them to build on their own resources. They empower parents and give them confidence in their own ability. Parents are given practical advice in the home* (Social Worker).

*Families in rural areas benefit immensely from this type of service. People in cities have so much on their doorstep* (Community Nurse).

*I think there are many families who suffer when they don’t know how to deal with different disabilities* (Family centre manager).

### 3.3. Children Develop

The parents reported a variety of ways in which their child had developed, but a recurring phrase was an increase in their confidence along with improved communication and social skills:

*My child has become more independent, confident and has grown so much since the start of the service. She is more understanding and her attention and behaviour have improved* (22).

*I have seen my daughter make friends, gain confidence and enjoy going out and wanting to get involved in things* (29).

Various examples were given of the way particular children had improved:

*He’s a lot happier, handling boundaries better, hygiene better, more friendships, better self-esteem and better knowledge of what is right and wrong* (11).

*My son has become a wonderful boy and has listened to me more. He’s really good at speaking. more independent* (29).

*It exposes our children to ‘life’, especially in areas they find challenging. This in turn helps us as a family to cope with day-to-day activities* (22).

Parents attributed the children’s gains to the activities which the project staff conducted in the family home and community activities:

*He looked forward every week to getting one-to-one time with his support worker. This made him feel valued, loved and special* (1).

*During the year they enjoyed spending time with the leaders and other children they were with. They found this difficult at the beginning. but they are used to meeting these people now* (12).

The Trust staff also reported on the gains children had made through increased social contacts both in the home and when going to community events and places:

*Children’s behavioural issues are addressed, they have more social outlets, the inclusion aspect is good as it boosts their self-esteem.* (Social Worker).

### 3.4. Community Connections

Parents can feel isolated especially in more rural communities a point stressed by staff who had referred families to the project:

*There’s no support in my area for families, no Sure Start, no crèches, no transport. Parents don’t drive. Lot of boredom in the home for the child. (Support staff) come to the house and take them out.* (Community Nurse).

The parents too commented on the connections that their child had made with other people and their participation in activities.

*My child could attend activities when I was unable to drive* (5).

*My daughter is going to Gymnastics and ballet and I am going to start her in Rainbows after Easter as she is turning four* (2).

*Her confidence has definitely grew as she has joined the local football and homework club in the area* (16).

*(He’s) Joined more groups, gained confidence to get up and sing at a talent contest* (30).

*I am more aware of activities for children and organisations that run wellbeing courses* (25).

Likewise, some parents also felt more connected with other parents and their local community:

*I went to some parent mornings; it’s nice to meet other parents, talking to others who understand, non-judgmental and share things with you* (75).

*We are now involved in other groups and have met other parents at fundraisers and education nights* (25).

*Meetings with other parents provides parental contact and sharing of information* (18).

*Getting a break has been great and feeling part of a community. It’s always good to know there is support out there—even if don’t avail of it all of the time. It’s reassuring to know it’s there if you need it* (35).

*Brighter Futures helped me realise that I am not alone as a parent. It’s OK to open up and ask for help and not to be embarrassed about asking for it. I am now more connected to know other organisations that can help my family* (98).

Trust staff noted how the parent’s increased connections were due to the example they had been given by the service:

*Parents getting support from the Brighter Futures Staff and then also learning how to support each other* (Family Centre Manager).

### 3.5. Supportive Staff

All the parents spoke highly of the project staff and the emotional support they had received from them. The most common descriptions were supportive, friendly and reliable. Trusted relationships seemingly had been built with families:

*Excellent staff—so warm, helpful, supportive and encouraging* (9).

*Staff very friendly and caring and good with my child* (14).

*Good reliable and friendly staff that give fantastic support* (14).

*Everyone is so kind and helpful, always happy and approachable* (29).

*Support in my home by someone who listens: a friendly face* (4).

Parents appreciated the willingness of staff to be available when needed and to go beyond their role.

*Always there at the end of the phone; always there to help when I needed them* (4).

*Even the knowledge alone that there is a group willing to help and support you as a parent is very important* (17).

*An excellent service both for the individual child and family. Supportive in all areas, easy to talk to all staff who go beyond anything that their job entails* (22).

*I got her diagnosis—one phone call, that’s all. Positive Futures was the only help I had. I would not ring social services, I will ring (Staff member), she makes me feel I am not a nuisance* (63).

Parents also commented on the practical help provided by the project:

*The picture aids staff provided helped with communication and increased her independence in dressing* (I1).

*The staff helped me to purchase a washing machine and with managing my money* (I2).

*They provided training courses on challenging behaviour and creative play* (I5).

*The way the service found ways of working around Covid was amazing. Through the pandemic we still felt very connected* (99).

*I had an issue with the bus and school. If something’s wrong (staff member) will go and fight for your rights. That means a lot, as parents are not listened to* (72).

Trust staff also recognized the important role played by staff:

*Mothers have someone to talk to, a listening ear. The staff understand what parents are going through. They give them hope. They see the potential in their child* (Social Worker).

### 3.6. Improvements to the Project

Parents were also asked to comment on any changes they felt were needed to make the project better. The most common comment related to the project being available for more than the allotted 12 months of home support:

*I’d love it longer than a year. So sad to say goodbye but a huge thank you for making such a difference* (9).

*Extend it to being for two years, not just one* (18).

*Service finished just as the summer holidays started. Left with no respite all summer* (23).

*My daughter has remarked that she makes friends with other girls and then loses contact* (5).

However, one parent commented:

*I personally think you are doing enough with the limited resources that you have. Parents/carers have to take on more involvement* (25).

Similarly, a parent who had left the project a year ago commented:

*The family does not need further support, we have already benefited and are set up in a much better place* (I6).

A social worker also endorsed this view, arguing that a time-limited project was necessary in order to build parental self-reliance as well as giving more families the opportunity to participate in it. Nonetheless, more flexible time spent on the project by certain families could be considered as well as having some form of prioritisation of referrals to it.

Another issue for some parents was the consistency of staffing, which is possibly reflective of the personal relationship and trust that staff had built with parents:

*Consistency. I think (the same) key workers (sic) where possible should go out with child so that a consistent approach is applied on visits and a transfer of consistency from home through to keyworkers so that ongoing progress within family unit isn’t lost* (21).

Other suggestions tended to relate more to individual family circumstances:

*Provision of car seats for visits (out of home) would be good* (1).

*Perhaps staff to receive training in peg feeding* (16).

*Would be good if Brighter Futures were provided with a wheelchair accessible vehicle to transport children and give their fulltime carers a break* (26).

*Maybe send photos or feedback from the two hours out with (the staff) to show what has been done and to show the various activities* (42).

The Trust staff spoke of the need for the project to be extended to other families in the county, especially as the waiting list for the project had to be closed for a time. They noted that there were few alternatives to which Trust staff could refer families. In addition, other projects work only with children, not parents and families. The extension of the project to include families awaiting a diagnosis was welcome.

A few suggestions were made by Trust informants for how some form of continuing support could be made available to families and the children once their time on the project is completed. These focused on strengthening and extending community linkages as a form of continuing support to families:

*Build up resources in the community—clubs and activities—that are geared up to take children and disabilities. This would be a good legacy from the project*.

*Continue with the Family Fun Days—make them open to all past as well as present families*.


*It helps when families know that there is someone they can call on if they have any questions, it helps when it is someone they trust and who is also familiar with the family.*


*Families can be re-referred back to the project should a particular need arise*.

*Train parents to be ‘trainers’ of others*.

The charitable funding for the project was available for five years. However, these respondents despaired of replacement funds coming from Health and Social Care services to enable the project to continue, to provide ongoing support to families once they leave the project, to enable increased number of families to participate in it or to instigate new developments such as supporting community initiatives. The available finances were directed towards families with critical needs.

After extensive lobbying, some funding is being made from statutory services, but it has meant scaling back the work of the project in the county.

## 4. Discussion

The Brighter Futures project confirmed the value of a family-centred support service for children with diagnosed or suspected developmental disabilities, such as autism and intellectual disability. However, more than this, the parental reports delineated the elements of the project that were especially pertinent to their own as well as their child’s needs, particularly for those families experiencing social deprivation and who have been less likely or able to engage with formal health and social services [21]. The challenge remains, however, of how to embed this learning into the delivery of the support services such as therapies, community nursing and special education that are commonly provided for children with disability or chronic health conditions.

The starting point for transforming these services into those aimed at strengthening families has been simply expressed as ‘think family’ [7,16]. The parents in this study endorsed this injunction as they emphasized the project’s engagement with the whole family and not just the identified child. This was achieved through visits to the family home, which take on extra significance in rural areas (where public transport is scarce), coupled with regular personal phone calls [22]. Ongoing conversations between the project staff and parents identified the learning targets most suited to the child and the activities that can be used at home to attain them. Shared decision-making helped to build parental confidence that they had the capacity to change and develop their child’s behaviours. Their reward was to see the children become more communicative and better at socializing, which were the most commonly identified concerns [16].

As the engagement with families deepened, parents could begin to appreciate how the relationships among family members—with siblings for example—could be changed to improve family functioning and achieve a better quality of family life [23]. The project assisted by including siblings in the home-based activities and organizing events for them outside the home to meet other siblings. Likewise, parent group meetings and occasional family days gave opportunities for fathers and mothers to meet others. In short, the project attended to the needs of the family and in so doing, benefited all its members and not just the child with additional needs.

Parents identified four outcomes they had experienced from the project. Although each outcome is presented separately, they interacted with each reinforcing one another (see Figure 1). First, the parents had become more confident in managing their child and felt better able to cope with challenges they encountered. Ironically, professional-led interventions can have the unintended consequence of undermining parents and devaluing their expertise [24], hence the project’s emphasis on consulting and listening to parents and also to the child with additional needs. When they take ownership of the intervention, the resulting gains can be attributed to their efforts. Opportunities to learn from other parents through informal mentoring was also a feature of the project’s approach, as previous research justifies [25]. Increased confidence brings with it wider personal gains, which was reflected in improved parental ratings of their emotional wellbeing and their participation in further education and social activities [16].

A second theme was the change they had witnessed in their child. Staff used their expertise to assess the functioning level of the child and attuned the chosen learning targets to it. In many instances, the selected targets focused on social and communication skills, on activities of daily life, such as dressing, crossing roads, and on managing unacceptable behaviours. Moreover, these changes were needed in the family context, yet they were unlikely to feature in the objectives chosen by therapists or teachers in clinics or school classrooms, and if even they did, the child’s learning had to generalize to the home. By contrast, embedding the learning activities in ‘real-life’ settings is a more efficient approach [26].

Community connections were also a feature of the project when project staff undertook learning activities in community facilities or going to and from them. This included local playparks, swimming pools, cafes and supermarkets. This gave children opportunities to develop their social and communication skills with peers and other adults in real-life settings rarely used by therapists or teachers. Parents who were hesitant to take their child out of the home for fear of them running away or having meltdowns, acquired the strategies and confidence to include the child in community outings. Likewise, parents and siblings were linked into existing community activities as a means of reducing their social isolation. Additionally, social activities were organized for families to meet one another in settings such as leisure centres. The intention was to help families to build social networks that could continue beyond the project [27].

The fourth theme was the support provided to the family by project staff. This included informational, tangible and emotional supports provided to the parents as well as to the child [28]. As noted above, during the project, the staff underwent training in supporting children and families and were provided with regular opportunities to share their experiences and expertise with one another. Even so, two further aspects were arguably crucial to their role. First, they were mostly recruited from the county and hence were familiar with its culture and customs and also had knowledge of the available facilities. Second, their personal qualities were well suited to the role: they were outgoing, friendly, welcoming, good listeners and flexible. Prospective staff perhaps can be mentored in all these qualities but they cannot be trained using traditional means. Rather, service managers need to recruit personnel who already have these attributes, which is challenging, especially in rural areas with smaller populations and the drift of talented young persons to the cities. One option worth considering is whether certain parents who have participated in the project could be recruited as staff members [29]. 

The study had a number of strengths. It drew on the experiences of over 60 parents who had taken part in a carefully designed project developed by an NGO with experience of family-centred practice. It took place in a rural county with elevated levels of social deprivation. The information was gathered rigorously and cross-checked. Many of the themes previously identified in the literature were now brought together in a conceptual model of family-centred support from the prospective and lived experiences of parents who had taken part.

However, the study had several limitations that relate mainly to the project. The parents participating in the project and in this study are possibly not representative of all parents of children with developmental disabilities. Some may be content with a child-centred focus and want to protect their family’s privacy. Ideally, families will be provided with options, and even family-centred projects need to be sensitive to the wishes of certain families.

A further limitation was the time-limited nature of the support provided to families. This resulted from the need to balance available staff resources with giving more families the opportunity of participating by creating an annual turnover. It is evident that certain families not only wanted but needed continuing support. Moreover, the analysis of quantitative data gathered by the project has identified the characteristics of families who may benefit from extended support, such as single parents in low-income households [16]. To an extent, this need could be met if some families were to receive less than 12 months home support so that others received more. It could also be met by creating other forms of ‘step-down’ support such as self-help groups or parent-led activities [7]. The bigger lesson, however, is that family-centred interventions need to be planned as part of a network of formal and informal supports and not just as a stand-alone service which frequently occurs. An unresolved question is which agency can take on this responsibility and if there is none, whether a multi-agency coalition of providers is a viable and cost-effective approach [30].

Future research should establish the replicability of these themes in other family-centred interventions with carers whose relatives have mental ill-health or dementia. Further follow-up of families who received family-centred support would help to determine its longer term impact and possibly the reduced demands that were placed on other health and social services. Such information would enable an assessment to be made as to the cost effectiveness of family-centred interventions [11].

## 5. Conclusions

Drawing on the lived experience of parents who had been involved in innovative support services, the core theme that emerged was the value of whole family engagement. Four inter-related subthemes were elaborated: parental confidence was boosted; children had developed; community connections were made and relations were built with supportive staff. These insights should help existing health and social care services to become more family-centred and inform the development of new support services in response to the high levels of unmet needs among marginalized families in even the most affluent countries.

## Figures and Tables

**Figure 1 ijerph-20-04205-f001:**
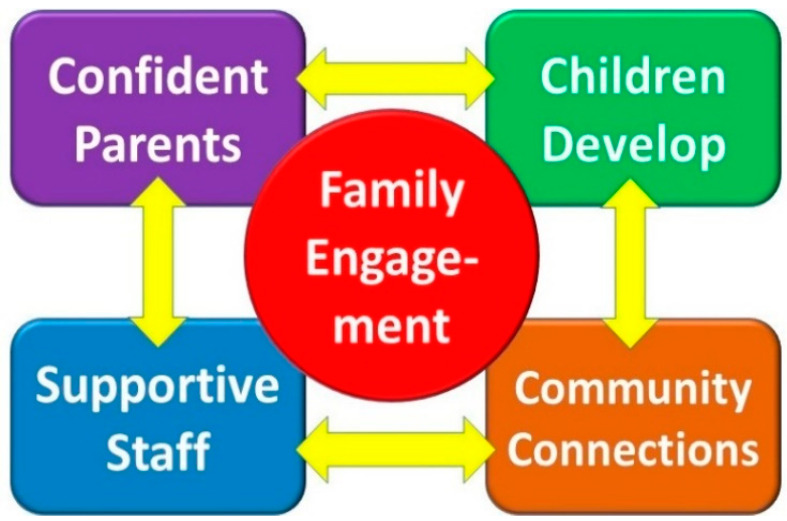
The themes identified in parents’ perceptions of family-centred support.

## Data Availability

The data reported in this paper is available on reasonable requests to the corresponding author.

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
