# Peer review of "Parental Perceptions of Family-Centred Supports for Children with Developmental Disabilities"

_ijerph, 2023, doi:10.3390/ijerph20054205_

Round 1
Reviewer 1 Report
Roy McConkey, Pauline O’Hagan, and Joanne Corcoran used a qualitative research methodology, conducting interviews with 16 parents who had been taken part in the study, with the aim of gaining insights into the ways a family-centred service had been of value to them. The study provided an opportunity to address more issues, especially for the rural community, documenting the lived experiences of families who had been involved in an innovative, family-centred support project for children with diagnosed or suspected developmental disabilities. Ideas for better meeting their needs were also sought. The project staff were recruited and lived locally, were familiar with the culture and customs of the area and knew the community facilities that were available. The staff had experience on working with children who had additional needs and received further training in family support and child development. They visited the family home monthly over a 12-month period with phone calls in between visits and identified in consultation with families, the learning targets that parents could focus on, they devised activities that could take place in the family home or during outings in the local community when the children could participate in leisure and sport activities, they also focused on the wellbeing of parents and provided a listening ear; directing parents to available community resources and accompanying them to clinic or hospital appointments, they organized social activities mainly for mothers but also for groups of siblings and fathers, arranged for families to meet each other socially.
The article gives an interesting scientific perspective on the gaps in the availability of high-quality evidence, especially for support services targeting parents having a child or more with a developmental or learning problem. The core theme that emerged was the value of whole family engagement. Four inter-related subthemes were elaborated: parental confidence had been boosted; children had developed; community connections were made, and the relations built with supportive staff. These insights should help existing health and social care services to become more family-centred and inform the development of new support services in response to the high levels of unmet needs among marginalized families.
The title and abstract are appropriate for the content of the text. Furthermore, the article is well constructed, the design and analysis were well performed, and the result are clearly presented. The conclusions drawn are adequately supported by the results.
I was wondering if the authors could develop a bit more the variety of intervention that were applied in relation with each type of beneficiary (parent, child, sibling), because it would be helpful for those interested in developing this type of service to know what type of training the staff will require, beside the personal abilities discussed in the article.
I recommend the article for publication.
Author Response
The article gives an interesting scientific perspective on the gaps in the availability of high-quality evidence, especially for support services targeting parents having a child or more with a developmental or learning problem. The core theme that emerged was the value of whole family engagement. Four inter-related subthemes were elaborated: parental confidence had been boosted; children had developed; community connections were made, and the relations built with supportive staff. These insights should help existing health and social care services to become more family-centred and inform the development of new support services in response to the high levels of unmet needs among marginalized families.
Many thanks for your summary of the article and for confirming the themes we identified with which you seem to concur after your reading of the quotations we presented in the article. We are pleased that you could see the applicability of the insights gained to wider service provision especially for marginalised families.
The title and abstract are appropriate for the content of the text. Furthermore, the article is well constructed, the design and analysis were well performed, and the result are clearly presented. The conclusions drawn are adequately supported by the results.
Our thanks again for this feedback. In particular my colleagues who designed the service were foremost social work practitioners rather than researchers so they are greatly heartened by the recognition you have given to the standard of their reporting.
I was wondering if the authors could develop a bit more the variety of intervention that were applied in relation with each type of beneficiary (parent, child, sibling), because it would be helpful for those interested in developing this type of service to know what type of training the staff will require, beside the personal abilities discussed in the article.
We are delighted to have the opportunity to provide the additional information you requested and we agree that this will further assist other service personnel and reseachers wishing to learn from our experiences and especially with respect to the training and guidance provided to front-line staff. We have provided additional content in lines 97 to 127 (372 words) and also provided a reference to an accompanying article that provides further details.
I recommend the article for publication.
Many thanks for your recommendation that greatly encourages us in our work. We appreciate the time you have given to reading and commenting on our study.
Reviewer 2 Report
This is a nice manuscript on a family centered support program. The article is clearly written. The methods are clear.
My main concern with this manuscript is in regards to the results section. While the results of the manuscript are provided clearly and are interesting, I feel that they are very much project specific. I did not feel that the quotes provided fed into higher thematic understanding. In some places it felt that there were too many short quotes that looked a bit like a list of quotes. That being said, I wonder if the results are relevant to the specific project and service and less to a wider audience.
Minor issues:
Line 110 - “Using local area Multiple Indicators of Social Deprivation for N.I. [17], nearly two-thirds of families (65%) resided in areas that fell within the top 30% of more deprived areas with only 3% living in the 30% least deprived areas.” – This sentence was not clear. Were any living in non-deprived area?
Line 129 – “(Some referred families had two or three children.)” – May be important to stress two or three children with disabilities.
Author Response
This is a nice manuscript on a family centered support program. The article is clearly written. The methods are clear.
Many thanks for reading our article and for your confirmation that the article is clearly written and that the methods are clear. Although in response to Reviewer 1 we have provided further examples of the learning targets selected with and for families.
My main concern with this manuscript is in regards to the results section. While the results of the manuscript are provided clearly and are interesting, I feel that they are very much project specific. I did not feel that the quotes provided fed into higher thematic understanding. In some places it felt that there were too many short quotes that looked a bit like a list of quotes. That being said, I wonder if the results are relevant to the specific project and service and less to a wider audience.
We are pleased that you felt the results had been presented clearly and that you found them interesting. The results indeed are project specific not least because we drew on parents lived experience of the project in which they had involved for around 12 months on a regular basis and some less regularly for a longer period of time. We were able to draw on parental reports gained both from interviews and written comments - the latter account for the shortened quotes. As the reviewer will appreciate, the main themes identified were further elaborated through the various examples that parents had provided which were of particular significance to them. We felt that these were best expressed in their own words rather than in synopsis that using our choice of words. We have added additional text to make this approach clearer at lines 201 to 205. Throughout the results section we have grouped the quotes under an introduction which means that no listing is longer than five quotations.
So we make no apology for the findings being project specific (which, we would contend, is often the case in much social service research projects and service evaluations).
The important, but different issue you raise as to the wider applicability to the findings to 'higher thematic understanding' we felt was best considered in the discussion but also using insights gained from our interviews with health and social care staff as noted in lines 190-192. Indeed most of the discussion section was given over to discussing the wider applicability of the themes - now highlighted in red - and we included citations to the extant literature in support of our contentions (which hopefully is what you meant by higher thematic understandings). Moreover we ended by noting how further research could add to these understandings. However we were reluctant cautious to over-generalise even though you kindly acknowledged the robustness of the study and the methodology we adopted.
Minor issues:
Line 110 - “Using local area Multiple Indicators of Social Deprivation for N.I. [17], nearly two-thirds of families (65%) resided in areas that fell within the top 30% of more deprived areas with only 3% living in the 30% least deprived areas.” – This sentence was not clear. Were any living in non-deprived area?
For purposes of identifying areas with higher levels of social deprivation, Northern Ireland is divided into nearly 800 small graphical areas that are then ranked in terms of highest to lowest levels of deprivation. No area is described as 'non-deprived', rather relative terminology used; i.e. least deprived. A reference to this methodology had been given We can add further clarification to the text if the editor deems it necessary.
Line 129 – “(Some referred families had two or three children.)” – May be important to stress two or three children with disabilities.
Thanks for this clarification that has been added at lines 152/153.
Round 2
Reviewer 2 Report
The article in its current format is suitable for publication.